# Antioxidant Therapy in Cancer: Rationale and Progress

**DOI:** 10.3390/antiox11061128

**Published:** 2022-06-08

**Authors:** Maochao Luo, Li Zhou, Zhao Huang, Bowen Li, Edouard C. Nice, Jia Xu, Canhua Huang

**Affiliations:** 1School of Medicine, Ningbo University, Ningbo 315211, China; 2020324060045@stu.scu.edu.cn; 2State Key Laboratory of Biotherapy and Cancer Center, West China Hospital, Sichuan University, and Collaborative Innovation Center for Biotherapy, Chengdu 610041, China; 2015224060079@stu.scu.edu.cn (L.Z.); huangzhao@scu.edu.cn (Z.H.); libowen@stu.scu.edu.cn (B.L.); 3Department of Biochemistry and Molecular Biology, Monash University, Clayton, VIC 3800, Australia; ed.nice@monash.edu

**Keywords:** reactive oxygen species, oxidative stress, antioxidants, cancer therapy

## Abstract

Cancer is characterized by increased oxidative stress, an imbalance between reactive oxygen species (ROS) and antioxidants. Enhanced ROS accumulation, as a result of metabolic disturbances and signaling aberrations, can promote carcinogenesis and malignant progression by inducing gene mutations and activating pro-oncogenic signaling, providing a possible rationale for targeting oxidative stress in cancer treatment. While numerous antioxidants have demonstrated therapeutic potential, their clinical efficacy in cancer remains unproven. Here, we review the rationale for, and recent advances in, pre-clinical and clinical research on antioxidant therapy in cancer, including targeting ROS with nonenzymatic antioxidants, such as NRF2 activators, vitamins, N-acetylcysteine and GSH esters, or targeting ROS with enzymatic antioxidants, such as NOX inhibitors and SOD mimics. In addition, we will offer insights into prospective therapeutic options for improving the effectiveness of antioxidant therapy, which may expand its applications in clinical cancer treatment.

## 1. Introduction

Redox homeostasis is essential for biological function and its disturbance leads to profound pathophysiological consequences in cells, which emphasize the balance between the relative abundance of reactive oxygen species (ROS) and antioxidants [1,2,3]. However, cells may generate excessive ROS as an unavoidable result of alterations in metabolic signaling pathways [4,5]. Oxidative stress arises when ROS are excessively produced, while antioxidants are relatively insufficient. The ROS levels are tightly regulated by antioxidant systems, including enzymatic antioxidant and nonenzymatic antioxidant systems. To accommodate oxidative stress, cells modify metabolic and genetic reprogramming, thereby leading to increased production of NADPH, glutathione (GSH, l-γ-glutamyl-l-cysteinyl-glycine), superoxide dismutases (SODs) and thioredoxins (TRXs), returning ROS to homeostatic levels [6,7,8].

When the high ROS level exceeds non-toxic doses, ROS may cause oxidative damage to macromolecules, such as nucleic acids, proteins, lipids and glucose, resulting in fragmentation of enzymes and structural proteins, membrane damage, gene mutations and even pro-oncogenic signaling activation [9,10]. Increased oxidative stress can initiate tumor development and contribute to tumor progression by directly oxidizing macromolecules or oxidative stress-caused aberrant redox signaling [11], demonstrating that high ROS levels may increase the risk of cancer when antioxidant systems are insufficient to protect cells from oxidative stress. Since oxidative stress plays an important role in carcinogenesis and cancer progression [2,12,13], it is an attractive idea to use antioxidants for the treatment of cancer. Numerous antioxidants were developed in the past few decades. They can be classified as nonenzymatic antioxidants, such as NF-E2 p45-related factor 2 (NRF2) activators [14], vitamins [15], *N*-acetylcysteine (NAC) and GSH esters [16,17], and enzymatic antioxidants, such as NADPH oxidase (NOX) inhibitors [18] and SOD mimics [19]. Some of them have shown potential to act as anticancer drugs and multiple antioxidant therapeutic strategies were explored in pre-clinical and clinical research [20].

In this review, we will summarize redox homeostasis mechanisms and the relationship between oxidative stress and cancer, providing a detailed description of the rationale for, and recent advances in, antioxidant therapy in cancer. In addition, we also highlight several kinds of antioxidant drugs in pre-clinical and clinical trials, discussing the promise and limitations of antioxidant therapeutic strategies in cancer.

## 2. Redox Homeostasis: The Biological Basis for Antioxidant Therapy

ROS are a class of highly reactive free radicals, such as hydroxyl radical (^•^OH), the superoxide radical (O_2_^•−^) and hydrogen peroxide (H_2_O_2_) [21,22]. The high intracellular ROS level-induced oxidative stress leads to the upregulation of antioxidant capacity to maintain redox homeostasis by metabolic rerouting or activation of genetic programs [23,24]. Disruption of redox homeostasis contributes to multiple human diseases, including cancer, and resetting redox homeostasis with antioxidants is a promising strategy to prevent tumorigenesis or inhibit cancer progression. It is well known that redox homeostasis is balanced by the equilibrium of ROS generation and ROS elimination. Therefore, we first describe the underlying mechanisms that regulate the cellular redox homeostasis (Figure 1).

### 2.1. Mechanisms in ROS Generation

ROS are prominently generated by transmembrane NOXs and other various oxidases from the mitochondrial electron transport chain (ETC) [25], endoplasmic reticulum (ER) [26] and peroxisomes [27], in response to intracellular signaling and extracellular stimuli. The mitochondrion functions as a highly dynamic organelle and an essential endogenous enzymatic source of ROS, which generates ROS through ETC, a series of electron transfer complexes located on the mitochondrial inner membrane [28,29]. The production of mitochondrial ROS is associated with the metabolism of glucose, fatty acids and amino acids (via glycolysis, β-oxidation and oxidative deamination, respectively), which provide precursors for tricarboxylic acid (TCA) cycle to produce metabolic substrates that enter the ETC [30,31]. In the mitochondrial ETC, ROS generation is probably due to the leak of electrons from complex I, II and III. During this process, oxygen is reduced with a single electron and thus generating O_2_^•−^, which can be dismutated to H_2_O_2_ [32,33]. The rate of ROS generation from the mitochondrial ETC is predominantly dependent on the concentration of the one-electron donor and the reaction rate between the donor and oxygen. The primary function of NOXs is to produce ROS, which is triggered by a variety of factors and reported to be associated with tumor development [34]. The NOX family consists of seven members, namely NOX1, NOX2, NOX3, NOX4, NOX5, DUOX1 and DUOX2 [35]. They catalyze the conversion of oxygen to O_2_^•−^ by transferring electrons to molecular oxygen in various subcellular compartments, such as the nucleus [36]. NOXs-derived ROS might activate the downstream secondary oxidase systems, such as xanthine oxidase and uncoupled endothelial nitric oxide synthase, further aggravating oxidative stress and accelerating the development of cancer [37]. ER is a protein-folding factory, which plays an important role in normal physiology [38,39]. The oxidizing site in ER supports the proper conformation and post-translational modifications of nascent proteins [40]. In response to the aggregation of unfolded or misfolded proteins within ER, glucose-regulated protein 78 (GRP78) dissociates from ER stress receptors, such as activating transcription factor 6 (ATF6), pancreatic ER kinase (PKR)-like ER kinase (PERK) and inositol-requiring enzyme 1 (IRE1), leading to ER stress and eventually resulting in ROS accumulation [41,42,43]. In addition, the release of calcium from the ER and depolarization of the mitochondrial inner membrane can stimulate the production of mitochondrial ROS and mediate excessive oxidative stress [44,45]. As multifunctional dynamic organelles, peroxisomes exist in almost all eukaryotic cells and play essential roles in redox homeostasis [46,47]. The name of peroxisomes derives from their function in the metabolism of H_2_O_2_ [27]. Peroxisomal respiration accounts for approximately 20% of total oxygen consumption and produces up to 35% of total H_2_O_2_ by peroxisomal oxidases in certain mammalian cells [48]. For instance, peroxisomal oxidase acyl-CoA oxidase 1 (ACOX1), the rate-limiting enzyme in fatty acid β-oxidation, can oxidize very long-chain fatty acid (VLCFA) and lead to H_2_O_2_ production in peroxisomes. In addition, a gain-of-function mutation in ACOX1 may further enhance the ROS levels [49]. Besides, the ACOX1-induction of ROS production was demonstrated to be involved in oxidative DNA damage and the progression of hepatocellular carcinoma (HCC) [50]. Ultraviolet (UV) radiation is also an important factor that contributes to ROS generation and subsequent carcinogenesis [51]. Cells exhibit an increased production of ROS when exposed to UV radiation. UV-induced transition-type mutations at dipyrimidine sites frequently occur in the RAS oncogene and p53 tumor suppressor gene [52]. In addition, a wide range of biological phenomena, such as inflammatory and oxidative modifications of macromolecules, were reported to participate in UV-induced skin carcinogenesis and the progression of glioblastoma [53,54].

### 2.2. ROS Elimination with Enzymatic or Nonenzymatic Antioxidant System

Increased accumulation of ROS can be eliminated by various enzymatic antioxidant systems including SODs [55], GSH peroxidases (GPXs) [56], peroxiredoxins (PRDXs) [57], paraoxonase (PONs) and catalase (CAT) [58]. Additionally, ROS can also be eliminated by nonenzymatic antioxidant systems, such as GSH [59] and TRXs [60]. The antioxidant systems counteract ROS-mediated damage to maintain ROS homeostasis, enabling tumor cell survival [20].

The enzymatic antioxidant system mainly consists of SODs, PRDXs, CAT, PONs and GPXs. Under oxidative stress, these antioxidant enzymes are upregulated or activated to prevent oxidative damage. SODs catalyze the conversion of O_2_^•−^ into molecular oxygen and H_2_O_2_, thus controlling the levels of ROS and limiting their potential toxicity [61]. Since SOD1 was firstly discovered in 1969, all of the three members in the SOD family were biochemically and molecularly characterized in mammalian cells, including Cu/Zn-SOD (SOD1), Mn-SOD (SOD2) and EC-SOD (SOD3) [62]. SOD1 and SOD2 localize in cytosol, the mitochondrial inter membrane space, the nucleus and the mitochondrial matrix, while SOD3 is secreted into the extracellular space [63,64,65]. The SOD family constitutes the first line of defense against ROS. The O_2_^•−^ is dismutated by SODs to form H_2_O_2_, which can be decomposed into O_2_ and H_2_O by CAT or GPXs [66]. Several enzymes, such as glutathione reductase and glucose-6-phosphate dehydrogenase, function as secondary antioxidant enzymes that enable GPX to function with cofactors (NADPH, GSH and glucose 6-phosphate) but not to act on ROS directly [67,68]. PON2 is one member of the PON family that consists of three members, namely PON1, PON2 and PON3. PON2 is a membrane-associated protein that is located in the plasma membrane, mitochondria and ER [69]. It was reported that PON2 protects against oxidative stress, both in vivo and in vitro [70,71]. For instance, PON2 binds with high affinity to coenzyme Q10 and protects against mitochondrial dysfunction when localized to the mitochondria, while PON2 deficiency results in mitochondrial oxidative stress [72].

Nonenzymatic antioxidants are non-catalytic small molecules that can quench ROS and reduce oxidative stress [73]. The most abundant nonenzymatic antioxidant is GSH, a tripeptide composed of glutamate, cysteine and glycine. Its synthesis is regulated by glutaminases (GLS1 and GLS2), the cystine-glutamate antiporter xCT (SLC7A11), the GSH biosynthetic rate-limiting enzyme glutamate-cysteine ligase (GCL) and the GSH synthetase (GSS) [74,75]. GCL is a heterodimeric holoenzyme that is composed of catalytic (GCLC) and modifier (GCLM) subunits; the expression levels of GCLC and GCLM are highly associated with the drug sensitivity of cancer cells and patient survival [76]. Moreover, the silencing of SLC7A11, GCLC and GSS represses the proliferation of clear cell renal cell carcinoma by decreasing the cellular GSH levels. However, reduced levels of GSH were also observed in patients with breast or colon cancers, especially in the advanced stages of these diseases, indicating the essential role of GSH in cancer cell survival [77,78]. Another nonenzymatic antioxidant is the TRX system, which is composed of TRXs and NADPH-dependent thioredoxin reductase (TrxR), which participate in the removal of harmful and excessive H_2_O_2_ [79]. There are two kinds of TRXs in mammalian cells, known as cytosolic TRX1 and mitochondrial TRX2 [80]. TRXs directly donate electrons to thiol-dependent PRDXs to remove H_2_O_2._ Oxidized TRXs are then reduced by TrxR, with NADPH as a cofactor [81]. Moreover, the oxidized PRDXs can also be reduced by TRXs [57]. Given the important role of the TRX system in cellular redox homeostasis, disturbance in the TRXs’ metabolism is highly associated with the progression and chemoresistance of multiple tumors [82], thus making TRXs essential targets for anticancer therapy.

## 3. ROS Promote Carcinogenesis and Cancer Progression

It was demonstrated that oxidative stress is involved in a wide range of pathologies including cancer, and increased production of ROS are common features of cancer cells. Although high ROS levels are cytotoxic and may exert anti-tumorigenic effects via oxidative damage and ROS-dependent death signaling, ROS play critical roles during tumorigenesis and cancer development. Here, we focus on the pro-tumorigenic role of ROS in malignant progression, which may be addressed with antioxidant therapy. The elevated levels of ROS from altered redox homeostasis contribute to the transformation of healthy cells into cancerous cells and enable their survival through two major mechanisms. The first is that ROS directly oxidize macromolecules, such as nucleic acids, proteins, lipids and glucose, resulting in gene mutation and aberrant inflammation [83]. The second mechanism involves oxidative stress-caused aberrant redox signaling. ROS, particularly H_2_O_2_ and O_2_^•−^, might function as signaling molecules to cause various signaling pathways to go awry and drive cancer progression [84,85] (Figure 2).

### 3.1. ROS-Mediated Oncogenic Mutations Promote Carcinogenesis

The elevated ROS level functions as a contributor to the malignant transformation of normal cells by inducing mutations in nuclear DNA (nDNA) or mitochondrial DNA (mtDNA), as well as by causing oxidative damage to biomolecules [86,87,88]. Excessive ROS are highly associated with both nDNA and mtDNA mutations, which were reported to result in aberrant inflammation and metabolism, thus promoting malignant transformation [89]. Overproduction of ROS causes nDNA mutation and genetic instability, which further activate multiple oncogenes and lead to abnormal metabolic activity and decreased antioxidant capacity. These events eventually promote the production of ROS in a positive feedback manner [90,91]. Increased ROS was demonstrated to promote chronic inflammation, one of the major causes of cancer, through inducing chemokines such as IL-8 and CXCR4, as well as inflammatory cytokines including IL-1, IL-6 and TNF-α [92,93]. In the context of cancer initiation, mtDNA is also an essential target of ROS, as mtDNA mutation was linked to carcinogenesis [94,95]. Each mitochondrion carries a few dozen mtDNA copies. Increased ROS-induced somatic mutations in mtDNA affect the function of ETC and the ATP synthase, which might promote a Warburg-like phenotype shift towards glycolysis. The metabolic shift can shape cell behavior and participate in oncogenic transformation in multiple types of cancer, such as colorectal cancer, lung cancer, gastric cancer, liver cancer and head and neck cancer [96].

### 3.2. ROS Function as Signaling Molecules to Drive Cancer Progression

In addition to supporting carcinogenesis, ROS were also demonstrated to sustain and accelerate cancer progression via epithelial-to-mesenchymal transition (EMT), which is involved in reprogramming the tumor microenvironment (TME) [97,98]. The TME is affected by ROS through regulating the function of T cells, tumor-associated macrophages (TAMs) and cancer-associated fibroblasts (CAFs) in TME [99]. The TAMs and CAFs promote cell proliferation, angiogenesis, immunosuppression and invasion, thus enabling cancer progression via the reciprocal crosstalk between cancer cells and the TME [100]. Moreover, regulatory T (T_reg_) cells and cytotoxic CD8^+^ T cells in TME can suppress effective tumor immunity and contribute to cancer progression, which is associated with poor response to immunotherapy [101,102]. In terms of the role of ROS in TME, H_2_O_2_ is thought to function as signaling molecules, which might cause metabolic changes in CAFs, such as altered glucose uptake and mitochondrial activity [103,104]. ROS also contribute to cancer progression by triggering the immunosuppressive properties of TAMs. For instance, mitochondrial ROS activate MAPK/ERK activity, which contributes to the secretion of TNF-α and subsequently promotes cancer invasion [105]. Furthermore, it was also demonstrated that O_2_^•−^ can suppress T cell-mediated inflammation, thus promoting TAM-mediated immunosuppression and leading to tumor development [106].

## 4. Antioxidant Therapeutic Strategies in Cancer

Given the important role of ROS in cancer, it follows that modulating ROS levels is a promising anticancer strategy. This may suppress ROS-induced carcinogenesis and cancer progression by inducing oxidative damage and ROS-dependent cell death [1,89]. Therefore, multiple antioxidants and weak pro-oxidants were explored in pre-clinical research and clinical evaluations. Cancer cells can produce excessive ROS through the above-mentioned mechanisms and increased formation of ROS are common features of cancer cells, which makes them more susceptible to a further increase in ROS than normal cells. Therefore, pro-oxidants may function as anticancer agents. For example, it was reported that exogeneous H_2_O_2_ can dramatically reduce the survival of MCF-7 cells with PRDX1 knockout, showing the potential of pro-oxidants to promote ROS-mediated cell death [107]. In addition, weak pro-oxidants may also function as important contributors to antioxidant therapy by boosting internal antioxidant capacity. However, treatment with weak pro-oxidants in cancer therapy still needs further investigation. Here, we focus on the antioxidant therapeutic strategies using antioxidants. Overall, antioxidant therapeutic strategies in cancer can be classified as targeting ROS with nonenzymatic antioxidants, including NRF2 activators [108], vitamins [109,110] (Figure 3) or targeting ROS with enzymatic antioxidants, including NOX inhibitors [18,111], SOD mimics [112], NAC and GSH esters (Figure 4) (Table 1) [113,114]. 

### 4.1. Targeting ROS with Nonenzymatic Antioxidants

The transcription factor NRF2 was considered as a master regulator of various homeostatic genes that defend against cellular stress, including oxidative stress [115]. Upon exposure to oxidative stress, the transcription factor NRF2 is released from its principal negative regulator Kelch-like ECH-associated protein 1 (KEAP1) and translocates to the nucleus, where NRF2 binds to antioxidant response element (ARE) and promotes the expression of antioxidant genes [116]. High expression of NRF2 was observed in various oxidative stress-related diseases including cancer, especially in NRF2-activated malignant tumors. NRF2 activators were considered as potential agents to prevent carcinogenesis or reverse cancer progression [117]. Five categories of NRF2 activator were developed, the underlying action mechanisms of which include: (1) modification on sensor cysteines of KEAP1, leading to the dissociation between NRF2 and KEAP1 [118,119]; (2) direct disruption of the KEAP1-NRF2 interaction [120]; (3) disruption of the interaction between NRF2 and β-transducin repeat-containing protein (βTrCP), which targets NRF2 for proteasome degradation [121]; (4) sequestration of KEAP1 into autophagosomes by p62 [122]; (5) upregulation of NRF2 protein levels by de novo synthesis that cannot be degraded by KEAP1 [123]; (6) inhibition of the NRF2 transcriptional repressor BTB domain and CNC homolog 1 (BACH1) [124].

The current development of NRF2 activators is mainly based on modifying sensor cysteines of KEAP1 and disrupting the KEAP1-NRF2 interaction. For instance, fumaric acid esters are oral analogs of fumarate that represent a group of NRF2 activators that work by modifying sensor cysteines of KEAP1, among which dimethyl fumarate (DMF) is the most successful example [125]. It was reported that DMF can alkylate Cys151 of KEAP1, leading to the dissociation of NRF2 and KEAP1 [126]. DMF metabolite monomethyl fumarate (MMF) was also demonstrated to react with KEAP1 through Cys151, thereby stabilizing and activating NRF2 [127]. DMF and its major metabolite MMF can reduce inflammatory responses and exhibit a favorable tolerability profile in clinical trials, showing promise for cancer treatment [128]. In addition, compounds that show improved bioavailability compared with MMF, through improving the release rate, were synthesized, such as TFM735, which is reported to activate NRF2 via the Cys151 in KEAP1, leading to the inhibition of IL-6 and IL-17 from peripheral blood mononuclear cells [129]. In addition, nitro fatty acids (NO2-FAs), such as nitro linoleic acid and nitro-oleic acid, are endogenous signaling mediators that react with Cys273 and Cys288 in KEAP1 through nitro alkylation, resulting in the activation of NRF2 and being implicated in anti-inflammatory activities [130]. Recently, the non-covalent NRF2 activators were developed, which directly disrupt the KEAP1–NRF2 protein–protein interaction via a cysteine-independent binding mechanism [131]. For instance, the bis-carboxylic acid compound CPUY192018 is a high-affinity KEAP1 ligand, which promotes the release of NRF2 from KEAP1 and enhances the expression of NRF2-target genes [132]. The sulfonamide-containing compounds were reported to inhibit the KEAP1–NRF2 interaction and enhance the expression of NAD(P)H: quinone oxidoreductase (NQO1), which reduces lung inflammation in animal models [133]. The naphthalene bis-sulfonamide was also reported to promote the expression of NRF2-target NQO1 and protect against dextran sulfate sodium (DSS)-induced colitis [134]. In addition to the above-mentioned compounds, (SRS)-5 and benzene-disulfonamides were also demonstrated to function as potent non-covalent NRF2 activators that disrupt the interaction between KEAP1 and NRF2 [135,136]. Altogether, these compounds are high-affinity ligands for KEAP1 and can directly block the KEAP1–NRF2 interface, thereby activating NRF2 downstream antioxidant genes and protecting cells from oxidative stress. Although current drugs mainly target KEAP1, it is noted that NRF2 might bind to ARE sequences in a KEAP1-independent manner, possibly involving the regulation of transcriptional repressor BACH1 [137]. Therefore, compounds that inhibit the binding of BACH1 to ARE-driven genes, such as HMOX1, were also developed [124]. Presently, more NRF2 activators eliciting beneficial effects are arising. However, treatment with NRF2 activators may inactivate drug-induced oxidative stress that normally would result in cell death. Therefore, it is necessary to monitor their clinical efficacy, given that the activation of NRF2 may contribute to the development of chemoresistance [138,139]. Taken together, NRF2 activators have shown potential for cancer therapy, but further investigations are also needed to demonstrate their clinical efficacy, especially in combination with chemotherapeutic drugs.

NAC is currently one of the most studied antioxidant agents that can be quickly absorbed via the anion exchange membrane and deacetylate to produce cysteine, thus replenishing GSH [140]. NAC can reduce cysteine conjugates and is used therapeutically for many human diseases, including cancers [141]. However, NAC was also reported to increase melanoma cell metastasis in NOD-SCID-*Il2rg*^−/−^ (NSG) mice [142]. GSH esters, the derivatives of GSH, were developed for GSH supplementation, since GSH cannot be effectively transported into cells and exogenously administered GSH is rapidly cleared in plasma. Ester derivatives of GSH, such as monoethyl (GSH-MEE), diethyl (GSH-DEE), monomethyl (GSH-OMe) and isopropyl esters have shown high efficiency in increasing cellular GSH level [143]. In addition, compared with oral administration, subcutaneous or intraperitoneal injection of GSH esters is more effective in elevating GSH levels in various tissues [144]. However, although the efficacy of GSH esters to alleviate oxidative stress in cells and animal models was demonstrated, clinical trials with GSH ester are still needed.

As the most widely used dietary antioxidants, L-ascorbic acid (vitamin C) and α- tocopherol (vitamin E) are of great interest in cancer therapy [145]. Vitamin C is a type of water-soluble vitamin that cannot be synthesized endogenously in the human body, but can only be provided by dietary supplement, making it an essential nutritional component [146]. Dehydroascorbic acid (DHA), the oxidized form of vitamin C, is absorbed from the renal tubules by renal epithelial cells and functions as a reductant and an enzyme cofactor [147]. It was described that high dose vitamin C shows promising antitumor efficacy in patients with advanced cancer [15,148,149,150]. However, the role of vitamin C in cancer treatment is still controversial, as half of the studies indicate that vitamin C has no significant effect on the incidence and mortality of cancer [151,152,153]. Vitamin E is lipid soluble and mainly localizes to the plasma membrane, where it functions as a ROS scavenger through reacting with free radicals, thus defending against oxidative stress [154]. It was reported that vitamin E only has low toxicity and causes no obvious side effects at high dose intake [155]. However, several animal studies showed that vitamin E supplements might promote carcinogenesis and cancer progression [156]. Overall, the controversial effect of antioxidants on cancer raises significant concerns regarding antioxidant supplements. Therefore, novel strategies are warranted to resolve the double-edged effect of supplemental antioxidants, including vitamin C and vitamin E.

### 4.2. Targeting ROS with Enzymatic Antioxidants

As mentioned above, the NOX family is a major source of ROS and excessive activation of NOXs can contribute to oxidative stress. Thus, agents that would efficaciously target NOXs to scavenge ROS might hold significant promise for cancer therapy [157]. There are two types of NOXs inhibitors, including peptidic inhibitors and small-molecule inhibitors, both of which are based on the mechanism of inhibiting NOX enzyme activity or suppressing the assembly of the NOX2 enzyme [158]. Small peptide inhibitors of NOX complexes have shown therapeutic potential. The first peptidic inhibitor is Nox2ds-tat ([H]-R-K-K-R-R-Q-R-R-R-C-S-T-R-I-R-R-Q-L-[NH2], also known as gp91ds-tat). Nox2ds-tat was reported to inhibit the assembly of NOX2, a complex that consists of six subunits: the Nox2 subunit (also known as gp91phox); p22phox, and four cytosolic components; p47phox (organizer subunit); p67phox (activator subunit); p40phox, and the small Rho-family GTP binding protein Rac1 or Rac2 [159,160]. Nox2ds-tat selectively blocks NOX2 activity through interrupting the Nox2–p47phox interaction [161]. The inhibitory effects of Nox2ds-tat were demonstrated both in vitro and in vivo. For instance, Nox2ds-tat was reported to inhibit the production of angiotensin II-induced O_2_^•−^ [162]. Moreover, administration of Nox2ds-tat by subcutaneous infusion significantly attenuated the production of vascular O_2_^•−^ and subsequent vascular inflammation in angiotensin II-infused rat model [34,163]. In summary, the viability of Nox2ds peptide as a NOX2 inhibitor was demonstrated, which is important for suppressing NOX2 activity and preventing excessive ROS production.

Currently, multiple small-molecule global inhibitors that inhibit NOXs or flavoproteins in general, were synthesized, including diphenyleneiodonium (DPI), ebselen and diapocynin [164]. Among them, DPI is the first identified and commonly used potential inhibitor of NOXs, which inhibits the production of ROS by forming adducts with FAD, potentially contributing to the reduction of ROS and showing anticancer properties in colon cancer cells [165]. However, as a nonselective inhibitor, DPI might target other flavin-dependent enzymes, such as xanthine oxidase and nitric oxide synthase. Ebselen and diapocynin are described as NOX inhibitors but were also previously found to display unrelated effects [166]. Unlike DPI, apocynin specifically prevents the activation of NOX2 by inhibiting the translocation of p47phox, thereby repressing the production of O_2_**^−^** in vitro and exhibiting anti-inflammatory activity in vivo [167]. In addition, other specific NOX inhibitors, were also identified via cellular and membrane assays [168]. For instance, fulvene-5, one of the fulvene derivatives that have a chemical similarity to DPI, could inhibit NOX2 and NOX4 in vitro, as well as block the growth of endothelial cell-derived neoplasia in mice [169]. However, despite the great efforts made by researchers, few NOXS inhibitors have yet reached clinical trials. It remains challenging to identify compounds that target NOX specifically and show a profound impact in alleviating cancer. Much more work is still needed to develop NOX inhibitors for the treatment of oxidative-stress-associated disorders, including cancer.

SOD is a metalloprotein that can efficiently eliminate O_2_^•−^ with a dismutation mechanism. SOD was developed as a drug known as orgotein, to defend against oxidative stress in mammalian cells [170]. The anti-inflammatory property of orgotein was demonstrated through preclinical and clinical studies [171]. It was also reported that orgotein can effectively prevent or reduce the side effects of radiation therapy in bladder cancer patients [172]. In addition, several types of SOD mimics were synthesized, such as metalloporphyrins, Mn (II) polyamines, Mn (III) salens, Mn (III) corroles and Mn (IV) biliverdins [173,174,175]. Although the rate constants are much lower than the enzymes, SOD mimics appear to be effective in extracellular fluids where the antioxidant enzymes are absent or at deficient concentrations [176]. Moreover, some SOD mimics may act as pro-oxidants rather than antioxidants, thereby activating rather than mimicking SOD [177].

Metalloporphyrins have emerged as the most studied SOD mimics, such as Mn porphyrins. Various Mn porphyrin compounds, including MnTM-2-pYp^5+^, MnTE-2-pYp^5+^ and MnTDE-2-ImP^5+^, have shown high SOD activity that dismutates O_2_^•−^ to H_2_O_2_ [178]. The protective and therapeutic potential of Mn porphyrins were demonstrated in animal models of diseases, including cancers. To date, more porphyrins or porphyrin-based SOD mimics were synthesized with the establishment of the structure–activity relationships between SOD and metal-site redox ability [19]. The Mn (II)-containing penta-aza macrocyclic manganese compound GC4419 (known as avasopasem manganese, AVA) was reported to enhance tumor-killing activity when synergized with radiation in head and neck cancer [179]. In addition, GC4419 can enhance the toxicity of high-dose vitamin C in a H₂O₂-dependent manner, promoting radiation-induced cancer cell killing [180]. Furthermore, GC4419 also exhibits therapeutic potential in the inflammation animal model [181]. Unlike GC4419, the Mn (III)- containing salen complexes, such as EUK-8, EUK-134 and EUK-189, are not specific and have dismutation activity on both O_2_^•−^ and H_2_O_2_, showing protective effects for various types of cancer [182].

In summary, multiple antioxidant therapeutic strategies were developed for cancer treatment, which can be classified into two different categories of groups according to their targets: enzymatic antioxidants and nonenzymatic antioxidants, both of which have shown potential to act as antioxidant drugs in pre-clinical and clinical research.

## 5. Perspectives and Conclusions

Because oxidative stress is a well-documented phenomenon in cancer, it is rational that antioxidants can significantly reduce cancer incidence and progression. Although multiple antioxidant therapeutic strategies were explored and some of them are undergoing clinical trial, their efficacy remains unsatisfied. The factors that impede the anticancer activity of antioxidants include: (1) most studies use pharmacological but not dietary doses based on in vitro studies, however, antioxidants may be affected by complex, in vivo conditions; (2) antioxidants might be distributed unevenly in different tissues, and probably cannot function due to the low bioavailability and bio-accessibility in some specific organ; (3) some antioxidants exhibit antioxidant or pro-oxidant properties depending on their concentration and the pressure of oxygen. These factors determine the distinct consequences of the supplementary antioxidants. Moreover, most chemotherapeutic drugs generate high levels of ROS and result in oxidative stress. Treatment with antioxidants in cancer patients might, therefore, lead to an antagonistic effect on chemotherapeutic drug-induced cell death.

As discussed above, multiple antioxidants failed to demonstrate efficacy in clinical practice. Given that most antioxidant capacity is attributed to endogenously antioxidant enzymes or antioxidants, we propose that treatment with weak pro-oxidants to boost antioxidant activity might be a promising way for cancer patients, although the underlying biological rationale warrants further investigation and long-term follow-up of interventions are needed. An improved understanding of these mechanisms will facilitate the development of novel therapeutic agents, which might be effective in the prevention and treatment of cancer.

## Figures and Tables

**Figure 1 antioxidants-11-01128-f001:**
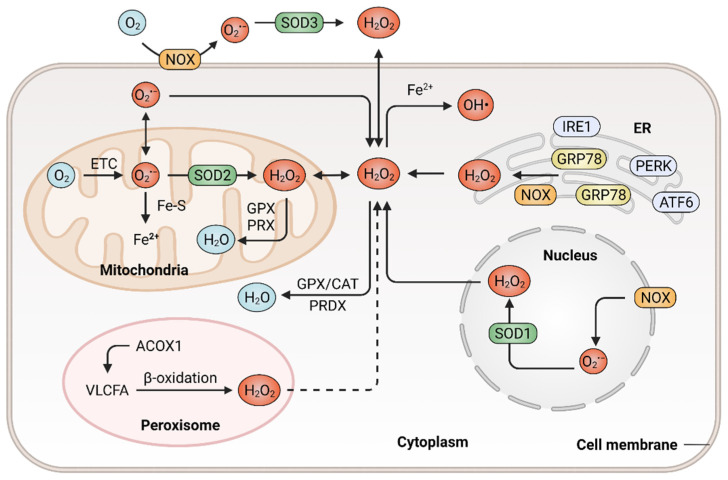
Generation and elimination of ROS in mammalian cells. ROS are generated extracellularly by NADPH oxidase (NOX) or intracellularly in different subcellular compartments, including endoplasmic reticulum (ER), peroxisome, nucleus as well as the mitochondrial electron transport chain (ETC). Antioxidant systems include the peroxiredoxin (PRDX), the glutathione peroxidase (GPX) and catalase (CAT) in the cytosol or mitochondria, which hydrolyze H_2_O_2_ to H_2_O.

**Figure 2 antioxidants-11-01128-f002:**
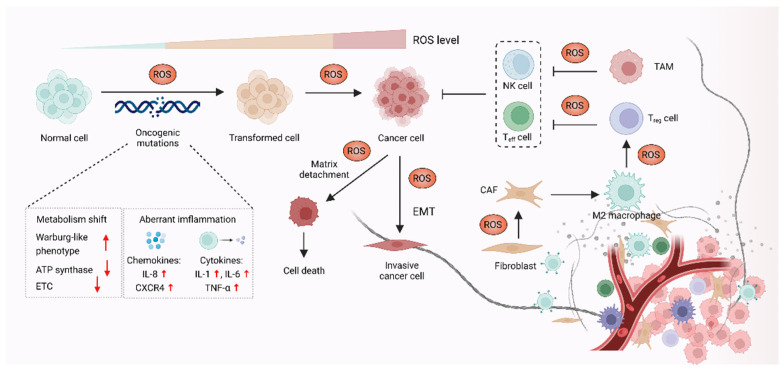
ROS promote carcinogenesis and malignant progression. In the process of carcinogenesis, ROS can contribute to DNA damage, which results in aberrant inflammation and metabolism, leading to oncogenic mutations and cell hyperproliferation. ROS can also act as signaling molecules to enable cancer cells’ survival and cancer progression via epithelial-to-mesenchymal transition (EMT). In addition, ROS might affect stromal cells, such as cancer-associated fibroblasts (CAFs), regulatory T (T_reg_) cells, effector T (T_eff_) cells and NK cells in the tumor microenvironment (TME) to promote cancer progression.

**Figure 3 antioxidants-11-01128-f003:**
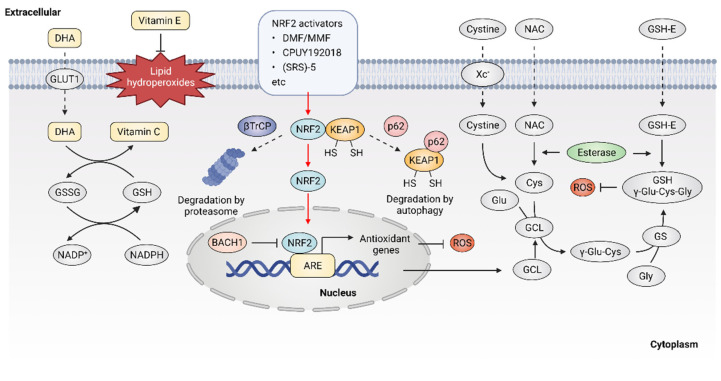
Targeting ROS with nonenzymatic antioxidants. Dehydroascorbic acid (DHA), the oxidized form of vitamin C, is taken up by cells through glucose transporter 1 (GLUT1) and then reduced to vitamin C. Vitamin E is located in cell membranes and defends against lipid hydroperoxides. NRF2 activators may disrupt the KEAP1-NRF2 interaction, leading to the activation of NRF2 downstream antioxidant genes. Glutathione (GSH) is synthesized from cysteine, glutamate and glycine. Exogenous N-Acetyl cysteine (NAC) and GSH esters (GSH-E) supplementation promote GSH production and defense against excessive ROS.

**Figure 4 antioxidants-11-01128-f004:**
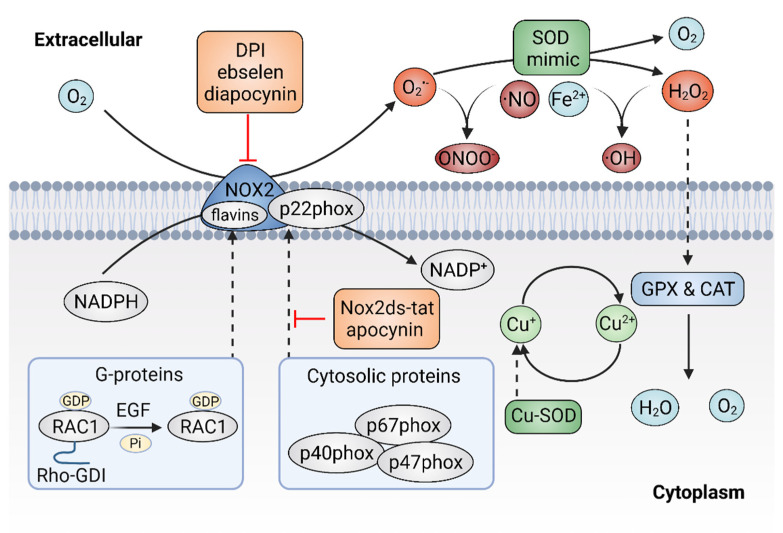
Targeting ROS with enzymatic antioxidants. The inhibitors of plasma membrane NADPH oxidase 2 (NOX2) can prevent the production of superoxide (O_2_^•−^) and superoxide dismutase (SOD) mimics might dismutate O_2_^•−^ to hydrogen peroxide (H_2_O_2_).

**Table 1 antioxidants-11-01128-t001:** Anticancer antioxidants in clinical trials.

Antioxidants	Cancer Types	Trial Status	Trial ID
NRF2 activators			
Sulforaphane	Lung cancer	Phase 2	NCT03232138
	Breast cancer	Phase 2	NCT00982319
	Prostate cancer	Phase 2	NCT01228084
	Colon cancer	NA	NCT01344330
	HNSCC	Early Phase 1	NCT03182959
Resveratrol	Colon cancer	Phase 1	NCT00256334
	Colorectal cancer	Phase 1	NCT00920803
	Neuroendocrine tumor	NA	NCT01476592
	Breast cancer	NA	NCT03482401
	Multiple myeloma	Phase 2	NCT00920556
Quercetin	Prostate cancer	Phase 1	NCT01912820
	Colorectal cancer	NA	NCT00003365
	Pancreatic cancer/NSCLC	Phase 2/3	NCT02195232
Curcumin	Breast cancer	Phase 2	NCT01042938
	Colorectal cancer	Phase 2	NCT02439385
	Prostate cancer	NA	NCT03211104
	Head and neck cancer	Early Phase 1	NCT01160302
	Pancreatic cancer	Phase 2	NCT00192842
Bardoxolone-methyl (CDDO-Me, RTA402)	Solid tumors/Lymphoid malignancies	Phase 1	NCT00529438
	Pancreatic cancer	Phase 1	NCT00529113
	Solid tumors/ Lymphoid malignancies	Phase 1	NCT00508807
RTA-408 (omaveloxolone)	NSCLC	Phase 1	NCT02029729
	Breast cancer	Phase 2	NCT02142959
	Melanoma	Phase 1/2	NCT02259231
Dimethyl fumarate	Multiple sclerosis	Phase 3	NCT02430532
	Lymphocytic leukemia	Phase 1	NCT02784834
	Glioblastoma	Phase 1	NCT02337426
Oltipraz	Lung cancer	Phase 1	NCT00006457
SOD mimics			
GC4419	Head and neck cancer	Phase 2	NCT04529850
	Pancreatic cancer	Phase 1/2	NCT03340974
	Squamous cell carcinoma	Phase 1	NCT01921426
	Head and neck cancer	Phase 2	NCT02508389
Metalloporphyrins	Lung cancer	Phase 3	NCT00054795
NOX inhibitors			
Ebselen (SPI-1005)	Cancer	Phase 1	NCT01452607
	Lung cancer, Head and neck cancer	Phase 2	NCT01451853
GSH-related antioxidants			
NAC	Breast cancer	Phase 1	NCT01878695
	Gastric cancer	NA	NCT03238404
	Ovarian cancer	NA	NCT03491033
	Head and neck cancer	Phase 2	NCT02123511
	Gastrointestinal neoplasms	Phase 2	NCT00196885
	Bladder cancer	NA	NCT02756637
	Lung cancer	Phase 2	NCT00691132
	Colorectal cancer	NA	NCT01325909
NOV-002	Breast cancer	Phase 2	NCT00499122
	Ovarian cancer	Phase 2	NCT00345540
	NSCLC	Phase 3	NCT00347412
	Leukemia	Phase 2	NCT00960726
Reduced GSH	Breast cancer	Phase 2	NCT00266331
Vitamins			
Vitamin C	Ovarian cancer	Phase 2	NCT00284427
	Pancreatic cancer	Phase 1	NCT00954525
	Prostatic neoplasms	Phase 2	NCT01080352
	Ovarian cancer	Phase 2	NCT00284427
	Advanced cancer	Phase 1/2	NCT01050621
	Solid cancers	Phase 1	NCT00441207
	NSCLC	Phase 1/2	NCT02655913
	Head and Neck Cancer	NA	NCT03531190
	Skin cancer	NA	NCT01032031
	Liver cancer	Phase 1/2	NCT01754987
Vitamin E	Prostate cancer	Phase 3	NCT00006392
	Colorectal cancer	Phase 1	NCT00905918
	Head and neck neoplasms	Phase 2	NCT02397486
	Skin neoplasms	NA	NCT02248584
	Pancreatic neoplasms	Phase 1	NCT00985777
	Breast cancer	Phase 2	NCT00022204

NA: Not Applicable; HNSCC, head and neck squamous cell carcinoma; NSCLC, Non-small cell lung cancer.

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
