# Peer review of "Antioxidant Therapy in Cancer: Rationale and Progress"

_antioxidants, 2022, doi:10.3390/antiox11061128_

Round 1

Reviewer 1 Report

The manuscript “Antioxidant therapy in cancer: rationale and progress” is a review regarding the recent advances in pre-clinical and clinical research on antioxidant therapy in cancer. The manuscript should be revised improving language and correcting some typos; some sentences are uncorrect or unclear (some of them are specified below). However, the manuscript is interesting for the readers, and could be considered for publication if authors will address the following issues:

  1. Please carefully revise the language of the manuscript; some sentences are wrong. For instance, in line 29 “Oxidative stress arises when are excessively produced while antioxidants are relatively insufficient”; probably authors meant Oxidative stress arises when ROS are excessively produced while antioxidants are relatively insufficient”. Please correct.
  2. Line 34: the word “etc” is inappropriate. Please specify all systems involved.
  3. Line 57 and everywhere else in the text: the superoxide anion must be written with the unpaired electron and negative charge as upperscript.
  4. In the paragraph 2.1 authors should discuss also the impact of UV radiation on ROS generation, since this is a crucial mechanism for carcinogenesis of tissues exposed to UV (PMID: 35453297).
  5. Figure 2: Please replace “ROS promotes” with “ROS promote”.
  6. In the paragraph 2.2 authors did not mention the enzyme paraoxonase-2, which exerts a potent antioxidant activity and was found to be upregulated in several malignancies, promoting cancer cell survival and chemoresistance (PMID: 33297311; PMID: 32093309).
  7. The authors are focused in supporting the idea that antioxidants are beneficial for cancer treatment. Indeed, they report many studies supporting this conclusion. However, a critical review must analyze also the studies that are not in accordance with this thesis. Piskounova et al. demonstrated that NOD-SCID-Il2rg(-/-) mice injected with the antioxidant N-acetyl-cysteine dramatically increased the metastasis formation, thus generating important questions regarding the use of antioxidants for cancer treatment (PMID: 26466563). Please discuss this theme through the text.
  8. Paragraph 4.1.” Targeting ROS with nonenzymatic antioxidants”: Although authors discuss the role of NRF2 as target of antioxidants compounds in cancer, it must be pointed out that in case of patients with chemoresistance, increasing NRF2 activity could worsen the clinical outcome of these patients since NRF2 plays a key role in cancer chemoresistance and its expression is increased in these patients (PMID: 35453348, PMID: 33805928). In fact, treatment with compounds activating NRF2 would inactivate oxidative stress drug-induced (especially in case of platinum-derived drugs) leading or worsen chemoresistance. This is a very important point of this manuscript that authors should discuss.

Author Response

The manuscript “Antioxidant therapy in cancer: rationale and progress” is a review regarding the recent advances in pre-clinical and clinical research on antioxidant therapy in cancer. The manuscript should be revised improving language and correcting some typos; some sentences are uncorrect or unclear (some of them are specified below). However, the manuscript is interesting for the readers, and could be considered for publication if authors will address the following issues:

1. Please carefully revise the language of the manuscript; some sentences are wrong. For instance, in line 29 “Oxidative stress arises when are excessively produced while antioxidants are relatively insufficient”; probably authors meant Oxidative stress arises when ROS are excessively produced while antioxidants are relatively insufficient”. Please correct.
Response: We are grateful that you raised this important point. As suggested, we have carefully read through the manuscript and have revised the language throughout the revised manuscript.

2. Line 34: the word “etc” is inappropriate. Please specify all systems involved.
Response: Thank you for your valuable comments. Following your suggestion, we have specified all systems involved (Please see Line 33-36 in the revised manuscript).

3. Line 57 and everywhere else in the text: the superoxide anion must be written with the unpaired electron and negative charge as upperscript.

Response: Thank you again for your valuable comment. We have carefully read through the manuscript and corrected the superoxide anion notation as requested (Please see Line 60, 80, 86, 125, 132, 182, 219, 363, 365 379, 388, 401 and 412 in the revised manuscript).

4. In the paragraph 2.1 authors should discuss also the impact of UV radiation on ROS generation, since this is a crucial mechanism for carcinogenesis of tissues exposed to UV (PMID: 35453297).
Response: Thank you for your valuable suggestion. We have discussed the impact of UV radiation on ROS generation and carcinogenesis as requested (Please see Line 109-115 and Ref 52-55 in the revised manuscript).

5. Figure 2: Please replace “ROS promotes” with “ROS promote”.
Response: Thank you very much for your comments. As suggested, we have replaced “ROS promotes” with “ROS promote” in figure 2 and carefully checked the language in the revised manuscript.

6. In the paragraph 2.2 authors did not mention the enzyme paraoxonase-2, which exerts a potent antioxidant activity and was found to be upregulated in several malignancies, promoting cancer cell survival and chemoresistance (PMID: 33297311; PMID: 32093309).
Response: Following your suggestion, we have discussed the antioxidant activity of paraoxonase-2 in paragraph 2.2 (Please see Line 136-141 and Ref 70-73 in the revised manuscript).

7. The authors are focused in supporting the idea that antioxidants are beneficial for cancer treatment. Indeed, they report many studies supporting this conclusion. However, a critical review must analyze also the studies that are not in accordance with this thesis. Piskounova et al. demonstrated that NOD-SCID-Il2rg(-/-) mice injected with the antioxidant N-acetyl-cysteine dramatically increased the metastasis formation, thus generating important questions regarding the use of antioxidants for cancer treatment (PMID: 26466563). Please discuss this theme through the text.

Response: Thank you very much for your considered comments. As suggested, we have discussed the potential unfavorable effects of antioxidants in cancer therapy, including the role of N-acetyl-cysteine in increasing the metastasis formation of human melanomas in NOD-SCID-Il2rg(-/-) mice (Please see Line 310-311 and Ref 142 in the revised manuscript).

8. Paragraph 4.1.” Targeting ROS with nonenzymatic antioxidants”: Although authors discuss the role of NRF2 as target of antioxidants compounds in cancer, it must be pointed out that in case of patients with chemoresistance, increasing NRF2 activity could worsen the clinical outcome of these patients since NRF2 plays a key role in cancer chemoresistance and its expression is increased in these patients (PMID: 35453348, PMID: 33805928). In fact, treatment with compounds activating NRF2 would inactivate oxidative stress drug-induced (especially in case of platinum-derived drugs) leading or worsen chemoresistance. This is a very important point of this manuscript that authors should discuss.
Response: Thank you very much for your valuable comments. Following your suggestion, we have discussed how the activation of NRF2 may contribute to the development of chemoresistance by inactivating drug-induced oxidative stress (Please see Line 300-306 and Ref 138, 139 in the revised manuscript).

Reviewer 2 Report

The authors reviewed the rationale for and recent advances in pre-clinical and clinical research on antioxidant therapy in cancer CoQ10 treatment with both enzymatic and non-enzymatic antioxidants. Although oxidative stress is necessary for cancer progression, therapeutic strategies with antioxidants are not fully effective. Furthermore, supplementation of some antioxidants instead might promote carcinogenesis and cancer progression. Finally, they propose that treatment with weak pro-oxidants conversely might be a promising way for anticancer treatment, but further investigation and long-term follow-up are required.

The reviewer thinks that the review might be interesting. However, the outcomes were inconclusive. Significantly, some reference reports did not meet the authors’ rationales, such as the descriptions in lines 146-148 and 302-303. In addition, the proposal using weak pro-oxidants in the last sentences seemed to appear suddenly. Probably, they might expect a chemical hormesis effect by pro-oxidants to boost internal antioxidant capacity like the radiation hormesis effect; however, it must confuse and might mislead reading audiences. Therefore, the authors’ proposal of using the weak pro-oxidants for cancer treatment should explain more closely and more thoroughly by adding a section in the main text.

Minor comments for improvement of the manuscript

  1. Line 47, nonenzymatic must be enzymatic.

  1. Lines 57, 78, and elsewhere, O2・- should be O2・-.

  1. Line 112, thioredoxins do not have enzymatic activity. Although they are small proteins, they just have antioxidative activities like GSH.

  1. Lines 206-207, epithelial-to-mesenchymal transition (EMT), is not described in Figure 2. Please add it to Figure 2.

  1. Lines 146-148 and 275-287, why GSH esters, the absorbent form of GSHs, were considered anticancer therapy agents regardless of whether GSH is essential for cancer cell survival?

Author Response

The authors reviewed the rationale for and recent advances in pre-clinical and clinical research on antioxidant therapy in cancer treatment with both enzymatic and non-enzymatic antioxidants. Although oxidative stress is necessary for cancer progression, therapeutic strategies with antioxidants are not fully effective. Furthermore, supplementation of some antioxidants instead might promote carcinogenesis and cancer progression. Finally, they propose that treatment with weak pro-oxidants conversely might be a promising way for anticancer treatment, but further investigation and long-term follow-up are required. The reviewer thinks that the review might be interesting. However, the outcomes were inconclusive. Significantly, some reference reports did not meet the authors’ rationales, such as the descriptions in lines 146-148 and 302-303. In addition, the proposal using weak pro-oxidants in the last sentences seemed to appear suddenly. Probably, they might expect a chemical hormesis effect by pro-oxidants to boost internal antioxidant capacity like the radiation hormesis effect; however, it must confuse and might mislead reading audiences. Therefore, the authors’ proposal of using the weak pro-oxidants for cancer treatment should explain more closely and more thoroughly by adding a section in the main text.

Response: Thank you very much for your valuable and positive comments. Following your suggestion, we have further discussed the multifaceted role of GSH in cancer development, and we believe the related descriptions have improved the revised manuscript. (Please see Line 150-154 in the revised manuscript). In addition, we have added a section about using weak pro-oxidants for cancer treatment in the main text (line 230-242).

Minor comments for improvement of the manuscript

Line 47, nonenzymatic must be enzymatic.

Response: Thank you for raising this point. As suggested, we have carefully read through the manuscript and corrected the language in the revised manuscript.

Lines 57, 78, and elsewhere, O2・- should be O2・-.

Response: Following your suggestion, we have updated the superscripted unpaired electron and negative charge of the superoxide anion throughout the manuscript (Please see Line 60, 80, 86, 125, 132, 182, 219, 363, 365 379, 388, 401 and 412 in the revised manuscript).

Line 112, thioredoxins do not have enzymatic activity. Although they are small proteins, they just have antioxidative activities like GSH.

Response: Thank you very much for your valuable comments. We have reorganized the paper and now the thioredoxins-related description has been moved to the nonenzymatic section (Please see Line 120, 152-163 in the revised manuscript).

Lines 206-207, epithelial-to-mesenchymal transition (EMT), is not described in Figure 2. Please add it to Figure 2.

Response: Thank you for pointing this out. We have now described EMT in Figure 2.

Lines 146-148 and 275-287, why GSH esters, the absorbent form of GSHs, were considered anticancer therapy agents regardless of whether GSH is essential for cancer cell survival?

Response: Thank you for your considered comments. Following your suggestions, we have further discussed the multifaceted role of GSH in cancer development. We believe this has improved the revised manuscript (Please see Line 150-154 in the revised manuscript).

Reviewer 3 Report

This review collected relevant data on the role of antioxidant activity in cancer. The paper is well done, however, the subject matter is not very original as there are dozens of reviews published in the last two years alone.

Moreover, the issues regarding the capacity of many types of cancers to enhance the intrinsic antioxidant defenses, which make them dependent on the efficacy of a given ROS-detoxifying system, should be considered. In fact,  an attractive target for anticancer therapy is the use of ROS-generating agents (i.e., prooxidants). In addition, should be taken into consideration that many chemo drugs "kill" cancer cells by inducing ROS-mediated cell death and anti-oxidant can reduce chemo drug activities.

Author Response

This review collected relevant data on the role of antioxidant activity in cancer. The paper is well done; however, the subject matter is not very original as there are dozens of reviews published in the last two years alone.

Response: Thank you very much for your considered and positive comments. We have carefully read through the manuscript, and have made extensive revisions. We believe the revised version is more readable and contains novel content.

Moreover, the issues regarding the capacity of many types of cancers to enhance the intrinsic antioxidant defenses, which make them dependent on the efficacy of a given ROS-detoxifying system, should be considered. In fact, an attractive target for anticancer therapy is the use of ROS-generating agents (i.e., prooxidants). In addition, should be taken into consideration that many chemo drugs "kill" cancer cells by inducing ROS-mediated cell death and anti-oxidant can reduce chemo drug activities.

Response: The existence of intrinsic antioxidant defenses in cancer cells is indeed an important issue and needs to be further investigated. Given the controversial role of ROS in cancer development, sensitive new ways for monitoring ROS levels in clinical samples and detecting the efficacy of given antioxidants may be the way forward. Following your suggestion, we have revised the manuscript and further discussed the potential role of pro-oxidants in cancer therapy (Please see line 230-242) and the possibility that antioxidants reduce the activities of chemotherapeutic drugs by suppressing the drug-induced oxidative stress (Please see Line 300-306, 429-432).

Round 2

Reviewer 3 Report

The MS was improved as requested.